# Comparative Analysis of miRNA Abundance Revealed the Function of Vvi-miR828 in Fruit Coloring in Root Restriction Cultivation Grapevine (*Vitis vinifera* L.)

**DOI:** 10.3390/ijms20164058

**Published:** 2019-08-20

**Authors:** Qiuju Chen, Bohan Deng, Jie Gao, Zhongyang Zhao, Zili Chen, Shiren Song, Lei Wang, Liping Zhao, Wenping Xu, Caixi Zhang, Shiping Wang, Chao Ma

**Affiliations:** 1Department of Plant Science, School of Agriculture and Biology, Shanghai Jiao Tong University, Shanghai 200240, China; 2Institute of Agro-Food Science and Technology/Key Laboratory of Agro-Products Processing Technology of Shandong, Shandong Academy of Agricultural Sciences, Jinan 250100, China

**Keywords:** grapevine, microRNA, fruit development, anthocyanin, high-throughput sequencing

## Abstract

Root restriction cultivation leads to early maturation and quality improvement, especially in the anthocyanin content in grapevine. However, the molecular mechanisms that underlie these changes have not been thoroughly elucidated. In this study, four small RNA libraries were constructed, which included the green soft stage (GS) and ripe stage (RS) of ‘Muscat’ (*Vitis vinifera* L.) grape berries that were grown under root restriction (RR) and in traditional cultivation (no root restriction, CK). A total of 162 known miRNAs and 14 putative novel miRNAs were detected from the four small RNA libraries by high-throughput sequencing. An analysis of differentially expressed miRNAs (DEMs) revealed that 13 miRNAs exhibited significant differences in expression between RR and CK at the GS and RS stages, respectively. For different developmental stages of fruit, 23 and 34 miRNAs showed expression differences between the GS and RS stages in RR and CK, respectively. The expression patterns of the eight DEMs and their targets were verified by qRT-PCR, and the expression profiles of target genes were confirmed to be complementary to the corresponding miRNAs in RR and CK. The function of Vvi-miR828, which showed the down regulated expression in the RS stage under root restriction, was identified by gene transformation in Arabidopsis. The anthocyanin content significantly decreased in transgenic lines, which indicates the regulatory capacity of Vvi-miR828 in fruit coloration. The miRNA expression pattern comparison between RR and CK might provide a means of unraveling the miRNA-mediated molecular process regulating grape berry development under root restricted cultivation.

## 1. Introduction

In recent years, with the development of dense and dwarf culture, root-limited cultivation, which is also called root restriction, rooting zone restriction, or root confinement, has been widely applied in fruit production. This technique limits the roots of fruit trees in a certain space by using physical or ecological materials, which regulate vegetative and reproductive growth by adjusting the relationship between aboveground and underground growth. The growth and development of roots are restrained by limited space under root restriction cultivation, which makes the growth of ground part more likely to transform toward reproductive development. Many studies on carambola [1], grape [2,3,4], Ponkan mandarin [5], cherry [6], and apple [7] have shown that root restriction cultivation can restrain the growth of vegetative organs, such as branches and leaves, but can promote reproductive growth and improve fruit quality, such as increased flavor and content of soluble solids. However, the effect of root restriction cultivation on fruit quality has only been studied at the physiological and biochemical levels, and the molecular changes that are caused by root restriction have rarely been reported.

As an important endogenous small non-coding RNA, miRNAs play a crucial role in plant development by guiding mRNA degradation or translational repression in posttranscriptional silencing [8,9,10,11]. With the development of high-throughput sequencing technologies, an increasing number of miRNAs have been identified from different species, including many horticultural plants, such as apple [12], cucumber [13], sweet orange [14], pear [15], grape [16], and pomegranate [17]. The majority of the miRNAs that were initially isolated from model plant *Arabidopsis thaliana* were found to be evolutionarily conserved across plant species [18]. Later, many researchers suggested that the miRNA repertoire of plant species comprises a set of non-conserved miRNAs besides the conserved ones [19,20,21]. Generally, non-conserved miRNAs are expressed at a lower level than the conserved miRNAs. To date, 48,885 mature miRNA sequences have been identified from 271 plant species, as revealed from the miRNA sequence database (miRBase 22; http://microrna.sanger.ac.uk).

With an increasing number of miRNAs being discovered and their functions being increasingly studied, it has been found that miRNAs act as powerful endogenous regulators in many different pathways [21,22,23], such as in the development of roots [24,25,26,27], shoots [28,29], leaves [30,31,32], and flowers [9,33,34]. For example, many studies have found that miR156, miR164, and miR166 play important roles in regulating leaf development [35,36,37,38], and miR156, miR159, miR319, and miR172 are involved in flowering regulation and phase changes from vegetative growth to reproductive growth [36,39,40,41,42,43]. Recent studies have shown that plant miRNAs are involved in the regulation of anthocyanin biosynthesis. In *Arabidopsis thaliana*, miR156 increased the synthesis of anthocyanin by regulating *SPL9* (squamosa promoter-binding-like 9) [44]. In apple, miR156, miR828, and miR858 could bind to multiple MYBs and regulate the metabolism of anthocyanin [45]. Under root restriction cultivation, so far, no studies have reported which miRNAs cause the increased anthocyanin content in grapevines.

To date, miRNA databases that are related to fruit development have been established by high-throughput sequencing in many species, such as pomegranate, sweet orange, orange, and corn, and conserved and specific miRNAs affecting fruit development and quality in corresponding species have been found [14,17,46,47]. Although it has been reported that many conserved miRNAs and grape-specific and tissue-specific miRNAs were identified in four miRNA libraries that were constructed from young leaves, tendrils, whole inflorescences, and newly developed small fruits of grape [16], to the best of our knowledge, studies on miRNAs that play an important role in grape fruit development has not been reported. In addition, studies on some key coding RNAs that affect grape fruit quality under root restriction have been performed [48], but as a regulator in plant growth and development, how miRNA affects grape fruit development and quality under root restriction has not been reported to date.

In this study, miRNAs from two developmental stages grape berries under root restriction and traditional cultivations were analyzed by high-throughput sequencing. A comparative profile of the miRNAs between grape berries under different cultivations revealed a set of 17 known and four novel miRNAs showing significant expression differences. The annotation of the potential targets of differentially expressed miRNAs indicated that highly ranked genes were implicated in biological processes, including anthocyanin biosynthesis, pectin metabolism, and transcription regulation. The heterogeneous expression of Vvi-miR828 (Vvi-miR828-OE) showed a phenotype of lighter colored leaves, which was consistent with the phenotype of grape berries in the control. For the first time, our study analyzed how root restriction cultivation affected the quality and development of grape berries at the miRNA level, which further provides a theoretical basis for the application of root restriction cultivation in fruit trees.

## 2. Result

### 2.1. Measurement of Soluble Solids, Titratable Acid, pH, and Anthocyanin Content in Grape Berries

The fruits of ‘Muscat’ in the RR group and CK group were collected three weeks after flowering (6 June 2017), and the last samples were collected when the fruits were ripe (9 August 2017). By observing the changes in the whole development process of grape fruit, we found that the grape fruit in the RR group began coloring on 14 July, while the CK group remained in the green fruit stage. As the fruit ripens, the color of grape berries in the RR group was always heavier than that of the CK group (Figure 1).

With the development of grape fruits, there was no significant difference in single fruit weight and the vertical and transverse diameter between RR and CK (Figure 2A–C). This result indicated that root restriction cultivation did not affect the size and weight of grape fruit. However, the skin color of the grape berries was darker in RR, which suggested a distinct difference in fruit quality. Subsequently, the soluble solids and titratable acid content, anthocyanin content, and pH of the fruit were measured. The results showed that the soluble solids and anthocyanin content, pH, and sugar-acid ratio of fresh berries in RR were significantly higher than that of CK after 14 July (2017), and the titratable acid content was opposite (Figure 2D–H). The results revealed that root restriction cultivation could increase the fruit quality of grapevines.

### 2.2. Sequencing and Annotation of Grape Fruit Small RNAs

The quality determination results showed that 14 July (2017) is the critical time point for the phenotype and global gene expression changes during fruit development in the RR group. sRNA libraries were generated from fruit of RR and CK collected on 14 July (RR-1 and CK-1) and 9 August (RR-2 and CK-2) to study the effect of root restriction cultivation on fruit quality. The principal component analysis of sRNA expression and cluster analysis between samples showed good repeatability among the three biological replicates (Figure 3). Deep sequencing of small RNA libraries yielded 17,598,617 and 14,647,769 unfiltered reads from RR-1 and CK-1, respectively, and 19,266,999 and 15,372,781 unfiltered reads from RR-2 and CK-2, respectively (Appendix A). After discarding the low-quality sequence reads, 5′- and 3′- adaptor reads, insert null, poly A reads, and sequences shorter than 17 nucleotides in the four libraries, 1,481,416 (71.88%, RR-1), 1,713,227 (75.22%, CK-1), 838,852 (69.14%, RR-2), and 1,103,035 (71.38%, CK-2) of the unique reads could map to the grape genome (Appendix A). The distribution of unique sRNA lengths was similar between RR-1, CK-1, RR-2, and CK-2, and the majority of sRNAs (approximately 71%) were 21–24 nt in length, with 21 nt sRNA being the major peak, which accounted for approximately 31 to 33% of the sRNA libraries, followed by the 24- and 22-nt classes (Appendix A). 

Against the Rfam database (version 10.0), the results showed that 0.22% and 0.20% of the sequences for RR-1 and CK-1 and 1.20% and 0.67% of the sequences for RR-2 and CK-2, respectively, matched non-coding RNAs, including rRNA, tRNA, snRNA, and Cis-reg. The miRBase database (V21.0) was used to annotate known miRNAs and identified 0.01%, 0.01%, 0.04%, and 0.03% of the unique sequences belonging to miRNAs in RR-1, CK-1, RR-2, and CK-2, respectively (Appendix A). The classification distribution of total sRNA and unique sRNA are summarized in Appendix A. The overall distribution pattern was similar between RR-1, RR-2, and CK-2, and the majority of total sRNAs were rRNA, while in the CK-1 group, the number of miRNAs was the highest.

The base bias on the first position from the 5′ end and base bias on each position were analyzed and summarized in Appendix A, and the pattern was similar between RR-1, CK-1, RR-2, and CK-2. With the exception of 17 nt for 17 to 24 nt sRNAs, the base bias at the first position from the 5′ end had a strong preference for U, particularly 23 nt sRNAs (Appendix A). In addition, for nucleotide bias at each position of 24 nt miRNAs, nucleotide U was the most prevalent at the first and 23rd positions. The proportion with U was approximately 80% and 90%, respectively (Appendix A). These findings are consistent with previous reports [19,20,49] that most of the miRNAs start with a 5′ U, which is one of the classic features of miRNAs, also it is not the case for other miRNAs, for example, 24 nt long miRNAs (lmiRNAs) are loaded into AGO4 proteins and mainly start with a 5′ A [50].

### 2.3. Known and Novel miRNAs

A total of 162 known miRNAs that belong to 44 families were identified in grape berries (Appendix A). The identified miRNA families are conserved in various plants, such as miR156, miR159, miR166 and miR169 [51]. In these four libraries, Vvi-miR169 was the largest family with 23 members, followed by Vvi-miR395 with 13 members. Nine miRNA families and 16 miRNA families had only one member and two members, respectively (Appendix A).

The sRNA sequences were mapped to the grape genome to identify novel miRNAs. These analyses revealed that 14 sRNA sequences were perfectly matched and they could be folded into stem-loop structures (Appendix A). A search of complementary miRNA* sequences showed that all 14 candidate miRNAs were considered to be novel miRNAs (Appendix A).

The count of each miRNA was normalized to transcripts per million (TPM) to compare the miRNA abundance in different libraries. In the four libraries, there were significant differences in the expression abundance of each miRNA, ranging from 0 to 29,896,116.67 TPM, and Vvi-miR166 and Vvi-miR3623-5p, were the most overrepresented families (Appendix A). For novel miRNAs, in the RR-1 library, the top three miRNAs with the highest abundance were miRN12*, miRN02, and miRN09. These three novel miRNAs are also frequently represented in the CK-1 library, but the difference is that the miRN02 abundance ranked third after miRN12* and miRN09. In the RR-2 and CK-2 libraries, the top three novel miRNAs with the highest abundance were miRN09, miRN01, and miRN02, but miRN01 was the most abundant in the CK-2 library (Appendix A).

### 2.4. Differentially Expressed miRNAs Between RR and CK

Analysis of the detected miRNAs in the four databases showed that 105 miRNAs were common in the four databases, and there was no unique miRNA in the CK-2 library, while there was one unique miRNA in RR-1 and two unique miRNAs in CK-1 and RR-2, respectively (Appendix A). The frequency of each miRNA in the small RNA library could serve as an index for the estimation of the relative expression abundance of miRNAs. For the comparison of the RR-1 and CK-1 sRNA libraries, seven miRNAs were down-regulated and six miRNAs were up-regulated in the RR-1 library (Figure 4A). For the CK-2 and RR-2 sRNA library comparisons, five miRNAs were down-regulated and eight miRNAs were up-regulated in the RR-2 library (Figure 4B). The above results revealed that root restriction cultivation might affect the quality of grape fruit by affecting the expression level of miRNAs to a certain extent.

In addition, we also analyzed the differentially expressed miRNAs at different developmental stages of grape fruit under the same cultivation conditions. When compared with RR-1, 10 miRNAs were up-regulated, such as Vvi-miRN01 and Vvi-miR169t, and 13 miRNAs were down-regulated, such as Vvi-miR2111-3p and Vvi-miR399a/h in the RR-2 sRNA library (Figure 5A). However, as compared with the CK-1 library, 13 miRNAs were up-regulated, such as Vvi-miR169f/g and Vvi-miRN01, and 21 miRNAs were down-regulated, such as Vvi-miR167c and Vvi-miR828 in CK-2 (Figure 5B). These results indicated that miRNA regulated the development of grape fruit through different transcriptions at different developmental stages of fruit.

Six known and two novel miRNAs were selected to carry out the tail-tailed qRT-PCR assay to further analyze the function of miRNA in fruit development and confirm the expression difference of the miRNA. The results showed that these eight miRNAs had the same expression pattern in CK and RR as that from sequencing data, and the abundance of each miRNA varied with the development of grape fruits (Figure 6; Appendix A).

### 2.5. Annotation of Potential Targets of Differentially Expressed miRNAs

Previous studies have shown that miRNAs bind to their target mRNAs with a perfect or nearly perfect match in plants. There were many differentially expressed miRNAs between RR and CK, and these miRNAs may regulate different targets during grape fruit development and ripening. Many studies indicate that most of the targets of known miRNAs are conserved across plants (including grapevine) [52,53,54], such as *SBP/SLP* (squamosa promoter binding protein-like) (targets of miR156), *MYB* (targets of miR159 and miR319), *NF-YA* (nuclear transcription factor Y subunit) and *WD48* (targets of miR169 family), *AP2* (the APETALA transcription factor 2) (targets of miR172), and *ARF* (auxin response factor) (targets of miR160) (Appendix A).The common function of the encoded proteins is the regulation of plant growth and hormone signal transduction.

For the novel miRNAs, the target genes of differentially expressed novel miRNAs were predicted, including Vvi-miRN01, Vvi-miRN02, Vvi-miRN05, Vvi-miRN08, Vvi-miRN12, and Vvi-miRN13 (Appendix A), to better understand their functions in grape. The results showed that, as consistent with previous reports, most of the novel miRNA targets belong to plant-specific transcription factors (e.g., *SCL21* (scarecrow-like 21) and *SCL1* (scarecrow-like 1)), followed by regulators of metabolic processes (e.g., protein kinases, pectinesterase, and callose synthase). In addition, there were several other targets whose functions are unknown. We further analyzed the expression pattern of some differentially expressed miRNA targets at five time points of grape fruit development by qRT-PCR. The results showed that the expression profiles of most target genes were complementary to the profiles of miRNAs, while some targets were not (Figure 7).

Gene ontology (GO) categories were assigned to the putative targets of the differentially expressed miRNAs according to the method described by Morin et al. to evaluate the potential functions of these miRNA-target genes [55]. Figure 8 summarizes the categorization of miRNA-target genes according to cellular component, molecular function, and biological process. Based on biological processes, the target genes were related to 10 biological processes, and the most frequent term was the pattern specification process for RR-1 vs. CK-1 and RR-2 vs. CK-2 (Figure 8A,B). Categories that are based on molecular function revealed that the miRNA-target genes were classified into 10 categories, and the most common term was DNA binding (Figure 8A,B). Moreover, the cellular component analysis revealed that the differences lie in the high percentage of nucleus in GO term (Figure 8A,B).

In addition, GO analysis was also performed on the differentially expressed genes of RR-1 vs. RR-2 and CK-1 vs. CK-2, and Figure 9 summarizes the categorization of miRNA-target genes. Based on the biological process, the pattern specification process was the most overrepresented GO term. DNA binding was the most frequent term based on molecular function, and notable differences are observed in the high percentage of nucleus for cellular component (Figure 9A,B).

### 2.6. Phenotypic Characterization of Vvi-miR828-OE Lines

Root restriction cultivation could make grape fruits color earlier and increase the content of anthocyanin in berries. A heterologous gene expression assay was carried out to further verify the role of differentially expressed miRNAs. Vvi-mir828, which has been reported to be involved in the metabolism of anthocyanin [56], was selected and overexpressed in *Arabidopsis thaliana*. Vvi-miR828-OE lines did not differ from WT in plant height and flowering, while the rosette leaves of Vvi-miR828-OE lines were pale green, and those of WT were dark purple (Figure 10A). We investigated the expression of ath-miR828 and its targets *AtMYB82* and *AtMYB113* in Vvi-miR828-OE and WT by qRT-PCR. The results showed that the abundance of miR828 in transgenic lines was significantly higher than that of WT, and the expression of target *AtMYB113* was lower in Vvi-miR828-OE lines than in WT, while the expression of *AtMYB82* in transgenic lines was not significantly different from that of in the WT plants (Figure 10B), which suggested that miR828 was involved in the metabolism of anthocyanin, mainly by regulating the expression of *AtMYB113*, and further indicated that root restriction cultivation might limit the expression of Vvi-miR828, thereby increasing the expression of target genes and promoting the content of anthocyanin of fruits.

## 3. Discussion

As a new cultivation mode, root restriction has been widely used in grapevine production, which regulates the aboveground part growth and development by controlling the development of roots (underground part). The growth potential of the grape tree was limited, and the fruit quality was improved under root restriction. In this study, ‘Muscat’ grape berries that were cultivated under root restriction and non-root restriction were used as material, and multiple physiological indicators during the whole development of fruit were determined. The size of the fruits was not affected under root restriction cultivation, but the soluble solids content, pH value, and anthocyanin content were significantly higher than that of the control group, while the titratable acid content was significantly lower than that of the control group in the later stage of fruit development (Figure 2). These results indicate that root restriction may mainly affect sugar metabolism and anthocyanin related gene expression, which thereby affects the metabolism and accumulation of sugar and anthocyanin in fruits. 

As a key component of complex networks of genes regulatory pathways, miRNAs can not only control plant growth patterns, but also play an important role in environmental responses [57]. Although miRNAs in grapes have been identified by high-throughput sequencing, how miRNAs regulate fruit quality in root restriction cultivation has not been reported. In this study, we characterized the small RNA transcriptome of grape fruits under root restriction and control. With a deep sequencing strategy using the Illumina platform, this approach provides us not only a large number of conserved miRNAs and novel miRNAs, but also the differentially expressed miRNAs between root restriction and control firstly. By taking a broader view of the sRNA transcriptome of grape fruit, we noticed that the size distribution of small RNAs is quite similar to those published plant species, such as Arabidopsis [58], apple [12], and pomegranate [17], with the total percentage of 21–24 nt sRNAs being more than 65%. We also found that the 24 nt sRNAs were not the most abundant, but 21 nt sRNAs (Appendix A), which is consistent with previous sRNA sequencing results of grape [16] and *Pinuscontorta* [59]. These analyses indicated that the small RNA transcriptomes have both similarity and specificity across plant species.

Based on the four sRNA libraries, the Venn diagram analysis showed that the miRNAs that are common in the four libraries accounted for more than 90% of the total number of miRNAs detected, and there were few unique miRNAs in each library (Appendix A). In the RR-1 and CK-1 libraries, there were 13 differentially expressed miRNAs, such as Vvi-miR160 (Figure 4A), and 13 differentially expressed miRNAs in the RR-2 and CK-2 libraries, such as Vvi-miR156 (Figure 4B). The expression of some differentially expressed miRNAs with high abundance was detected by qRT-PCR analysis (Figure 6). It was speculated that root restriction cultivation might influence the expression level of miRNA, rather than inhibiting or activating the expression of miRNAs, to cause phenotypic changes in fruit.

In plants, miRNAs play regulatory roles by mediating the cleavage of target mRNAs or repressing the translation of target mRNAs [60]; therefore, the initial prediction of target genes is necessary to shed light on the putative function of miRNAs in plant species/tissues/grown stages. In this study, the target predictions of these differentially expressed miRNAs, including known and novel miRNAs, were predicted by psRNA-Target (the updated version of miRU) [61], and GO categories were assigned to the predicted genes (Figure 8 and Figure 9), which showed that the highly ranked target genes were implicated in two biological processes: pattern specification and regulation of transcription. It is easy to understand the high frequency of terms ‘regulation of transcription’, because miRNAs are involved in diverse regulatory events [62]. Although further direct validation is necessary to elucidate the true in vivo targets, the annotations of these predicted miRNA targets could provide an alternate view of the gene regulation of fruit quality information under root restriction. 

Anthocyanin accumulation is developmentally regulated by subspecialized R2R3-type MYB transcription factors. With the rapid development of molecular biology techniques, it has been found that miRNAs have an irreplaceable regulation of plant color formation at the posttranscriptional level. The expression of Pg-miR156/Pg-miR157 increases in mature pomegranate, along with anthocyanin accumulation [17]. *SPL* is abundant in the early stage of fruit development when there is an absence of miR156 expression [44]. MiR828 is a novel miRNA discovered by high-throughput sequencing in recent years, but its functional characteristics have not been fully determined to date. In Arabidopsis, the tasiRNA species TAS4-3′D4(−) are generated by miR828-mediated cleavage of *TAS4* (Trans acting small-interfering RNA locus4) [20] and negatively regulated *MYB*. *AtMYB113* is targeted by miR828 directly in addition to TAS4-3′D4(–). Yang et al. found that the anthocyanin content in wild-type plants was 2.5 times higher than that in the miR828-overexpressing lines in Arabidopsis [56], which indicates that miR828 inhibits the synthesis of anthocyanin in vivo. In this study, the abundance of Vvi-miR828 was different in the CK-1 and CK-2 database comparisons (Figure 5), and qRT-PCR results showed that the abundance of Vvi-miR828 under root restriction cultivation was lower than that of in CK (Figure 6). Many studies have shown that most of the targets of the known miRNAs are conserved across several plants [12,13,17,52]. It is speculated that Vvi-miR828 had a negative regulatory effect on the synthesis of anthocyanins, which further indicated that root restriction cultivation affected the expression of Vvi-miR828 and increased the content of anthocyanin in grape berries. The Vvi-miR828 heterogeneous expression assay confirmed this result. The transgenic lines showed no difference from WT in vegetative and reproductive growth, while the accumulation of anthocyanin in rosette leaves decreased (Figure 10A). 

Taken together, in this study, we used high-throughput sequencing technology to provide a large-scale analysis of miRNA in ‘Muscat’ grape fruit and found some differentially expressed miRNAs. We first analyzed the miRNAome and possible miRNA-directed molecular processes in fruit under root restriction and preliminarily explored the molecular mechanism of root restriction on improving fruit quality, which provided a theoretical basis for root restriction cultivation technology. Further studies should be designed and executed to discover the function of these differentially expressed miRNAs and their targets to gain a more comprehensive understanding of the effects of root restriction cultivation in grape fruit. 

## 4. Materials and Methods

### 4.1. Materials

Research was carried out with three-year-old table grape ‘Muscat’ (*Vitis Vinifera* L.) during the fruiting season of 2017and the trees were planted in the orchard at the School of Agriculture and Biology of Shanghai Jiao Tong University (Shanghai, China). The experiment was divided into two groups of 13 grape trees each: the grape trees planted in round plastic containers with a diameter of 80 cm and 80 cm high were the first group as the RR treatment, whereas those of another group were planted in raised beds with the same soil at the open field as the control group (CK). The same watering and fertilizer strategies were applied to RR and CK. Ten shoots were left in each tree, and they were pulled upward at a 60° angle. One ear with approximately 50–80 grains was left on one shoot. The samples were collected three weeks after flowering (6 June 2017), and the last sample was collected on August 9th. The samples were collected nine times. For each sample, four ears were randomly selected from one tree and 15 grains were taken from the upper, middle, and lower parts of the ear. Four trees were selected for each biological replicate, and three biological replicates were performed. Fifty grains of each biological replicate were randomly selected for measuring the weight, vertical diameter and transverse diameter of one grain, soluble solids content, titratable acid content, pH, and anthocyanin. The remaining parts were frozen in liquid nitrogen and then stored at −80 °C until further use.

### 4.2. Measurement of the Fruit Size

After each sample was collected, the vertical diameter and transverse diameter of the ‘Muscat’ fruits in the RR and CK groups were measured by the Verniercaliper (Guilin, Guangxi, China), and the weight of the grain was measured by the electronic balance with an accuracy of 0.0001 g.

### 4.3. Determination of Fruit Quality Index

Measurement of soluble solids content: After peeling the grains, the flesh was broken with the juicer and placed into centrifuge tubes and centrifuged at 4 °C and 10,000 rpm for 10 min. The supernatant was used to measure the total soluble solid (TSS) content with a Brix meter (Hangzhou, Zhejiang, China).

pH determination: The pH value of the juice was determined by the METTLER TOLEDO pH measuring equipment (Shanghai, China) and repeated three times.

Measurement of titratable acid (TA) content: The titratable acid content was determined by acid-base neutralization titration. Three milliliters of the above supernatant was accurately taken, distilled water was added to a volume of 50 mL, and the mixture was placed into a 150-mL conical glass bottle. Two drops of 1% phenolphthalein were added into the conical glass bottle. The mixture was titrated with calibrated NaOH solution until the mixture initially turned pink and it did not fade for half a minute, and the NaOH dosage was recorded. The titration was repeated three times, and the average was obtained, which was converted into tartaric acid equivalent. The total acidity is calculated by the following formula: total acidity (%) = *V*2 × *N* × conversion coefficient × 100/*V*1, and in the following formula:

*V*1—the volume of the sample solution taken before titration (mL); *V*2—the volume of NaOH standard solution (mL); *N*—molar concentration of NaOH standard solution (mol/L); and, Conversion coefficient—tartaric acid 0.07.

Determination of anthocyanin content: According to the method of Wang et al. [63], 30 grains were selected from each group and ground into power in liquid nitrogen. A power of 0.2 g was taken, placed into 10 mL of 1% HCl methanol for extraction at 4 °C for 4 h, and then centrifuged at 4 °C and 12,000 rpm for 10 min. The supernatant was measured for the light absorption values at 553 nm and 600 nm while using a spectrophotometer (Shanghai, China). The variation of optical density (OD553–OD600 nm = 0.01) of the extraction from per gram fresh fruit was as an anthocyanin unit and it is represented by U.

### 4.4. RNA Extraction and cDNA Synthesis

Total RNA was isolated from frozen powder of some whole grape grains while using a modified CTAB method [64] and was treated with DNase I (Invitrogen, California, CA, USA) to remove DNA contamination. RNA integrity was verified by electrophoresis on a 1.2% agar gel, and the RNA was quantified while using a NanoDrop 1000 (New Boston, Massachusetts, MA, USA) and used for small RNA sequencing and real-time PCR. Approximately 2 μg of RNA was used as template for first-strand cDNA synthesis using SuperScript reverse transcriptase (Invitrogen), according to the manufacturer’s instructions (Invitrogen) for analyzing the abundance of target mRNAs, and miRNA RT Enzyme Mix (Beijing, China) was used to synthesize cDNA for analyzing the abundance of miRNAs. Total RNA from each of the three biological replicates was independently used in qRT-PCR analysis.

### 4.5. Small RNA Library Construction and Sequencing

The construction of the sRNA library and sequencing consisted of the following steps [65]. Total RNA was extracted from the grapefruits at the green soft stage (5, RR-1 and CK-1) and the fully ripe stage (9, RR-2 and CK-2) in the root restriction group and the control group for small RNA sequencing. Low-molecular-weight RNA was enriched by 15% PAGE gel, and 5′ and 3′ adaptors were added and then amplified by RT-PCR while using adaptor-specific primers following the Illumina protocol. Sequencing was performed by the YauGui (Shanghai, China) Biotechnology Co., Ltd. with the IlluminaHisSeq 2000 platform (Shanghai, China), and three independent biological replicates per sample.

### 4.6. Bioinformatic Analysis of miRNAs

After sequencing, to identify the known and novel miRNAs in grapefruit, raw reads were filtered to remove low-quality sequences, adaptor sequences, reads < 18 nt length, and reads with poly N from the raw data by FastQC software. Usually, the sRNA measures 17 to 30 nt (miRNA, 21 or 22 nt; siRNA, 24 nt; and piRNA, 30 nt). All unique clean reads in each library, specifically non-redundant ones, were considered for further analysis, including noncoding RNA identification and proper annotation. The unique clean reads were searched against the Rfam database using the blastn software (version 10.0). The results with E-values≤ 0.01 were extracted. Unique reads that matched known plant structural RNAs (rRNAs, tRNAs, snRNAs, and snoRNAs) were removed from further consideration. Small RNA reads were then mapped to the grape genome database and the grape pre-miRNAs/miRNAs database in miRBase 21.0 by bowtie software. Three mismatches were allowed between the reads and the known pre-miRNA/miRNA sequences. The reads that were mapped to known pre-miRNAs/miRNAs and also mapped to the grape genome were identified as conserved miRNAs. In addition, we searched miRNA* sequences (complementary to miRNA in the precursor molecule) in the sRNA libraries, the reads that did not map to known pre-miRNAs/miRNAs but mapped to the grape genome, and only those with miRNA–miRNA* duplexes, were regarded as novel miRNAs. Furthermore, the secondary structures of all identified and potential pre-miRNAs in the grape genome were predicted by MFold software [66]. The minimal folding energy indexes (MFEIs) of the novel miRNAs should be equal to or greater than 0.9 [67,68,69].

### 4.7. Analysis of Differentially Expressed miRNAs

The differential expression of miRNA was calculated by the negative binomial distribution test calculation of the DESeq package (http:/biioconductor.org/packages/release/bioc/html/DESeq.html) [70], the number of reads was tested for significant difference by NB (negative binomial distribution test), and the expression of miRNAs was estimated by the base mean value. The miRNAs with a *p*-value < 0.05 and log_2_ (fold change) >1 or <−1 were considered to be up-regulated or down-regulated.

### 4.8. Prediction of Target Genes for Known and Novel mRNAs and GO Analyses

The target prediction of miRNAs from grape fruit was performed while using psRNA Target tools (A Plant Small RNA Target Analysis Server 2017 Update, http://plantgrn.noble.org/psRNATarget/) with the expectation value at 5.0 and seed region at 2–7 nt, which involved loading miRNA reads into a FASTA file format to search for known targets in the grape (*Vitis vinifera*) transcript database. Predicted target genes were selected for further validation and functional analysis while using BLASTN (http://blast.ncbi.nlm.nih.gov). The Gene Ontology (GO) database (http://www.geneontology.org/) was then searched to annotate the putative genes that are involved in cellular components, biological processes, and molecular functions. All of the predicted target genes were mapped to GO terms in the GO database by counting the percentage of gene numbers for each term. We used hypergeometric distribution text to calculate the *p*-value. GO terms with a *p*-value ≤ 0.05 were considered to be significantly enriched in the predicted target genes.

### 4.9. Quantitative RT-PCR (qRT-PCR) Analysis 

qRT-PCR was conducted in 10-μL reactions while using SYBR Green Supermix (Beijing, China) to analyze the abundance of target mRNAs and miRcute Plus miRNA Premix for analyzing the abundance of miRNAs. Appendix A lists the primers that were used for these reactions. The relative abundance of miRNAs and their target gene mRNAs were calculated by the 2^−∆∆*C*t^ method [71] and normalized by U6 and β-actinas references [72], respectively.

### 4.10. Plant Material and Growth Conditions

*Arabidopsis thaliana* Columbia (Col) was used as the wild type. The seeds were sown on a mixture of vermiculite and nutritive soil, and the plants were maintained at 21 °C with a 16 h light and 8 h dark photoperiod. 

### 4.11. Construction of Plant Expression Cassettes, Plant Transformation and Characterization of Transgenic Plants 

The pri-miR828 was amplified while using the primers that are listed in Appendix A and then cloned into the pCambia1300 vector. The construct was confirmed by DNA sequencing before being introduced into the *Agrobacterium tumefaciens* strain GV3101. The floral dipping method was used to create transgenic plants [73]. The transgenic seeds were screened on MS (Murashige and Skoog) plated containing 30 mg/L hygromcin. Four independent T1 lines for the construct were generated. Homozygous T3 seeds of six representative lines were used for phenotypic analysis. 

## Figures and Tables

**Figure 1 ijms-20-04058-f001:**
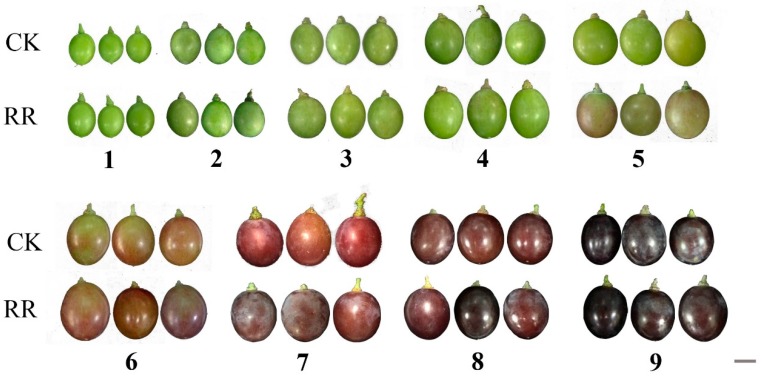
Morphological features of grape fruit development stages. Harvested fruit at different developmental stages from 21 days after pollination. RR: Root restriction, CK: control group. 1–9 indicate that the sample collection time is 6 June, 14 June, 22 June, 4 July, 14 July, 18 July, 23 July, 1 August, and 9 August 2017, respectively. Scale bar: 1 cm.

**Figure 2 ijms-20-04058-f002:**
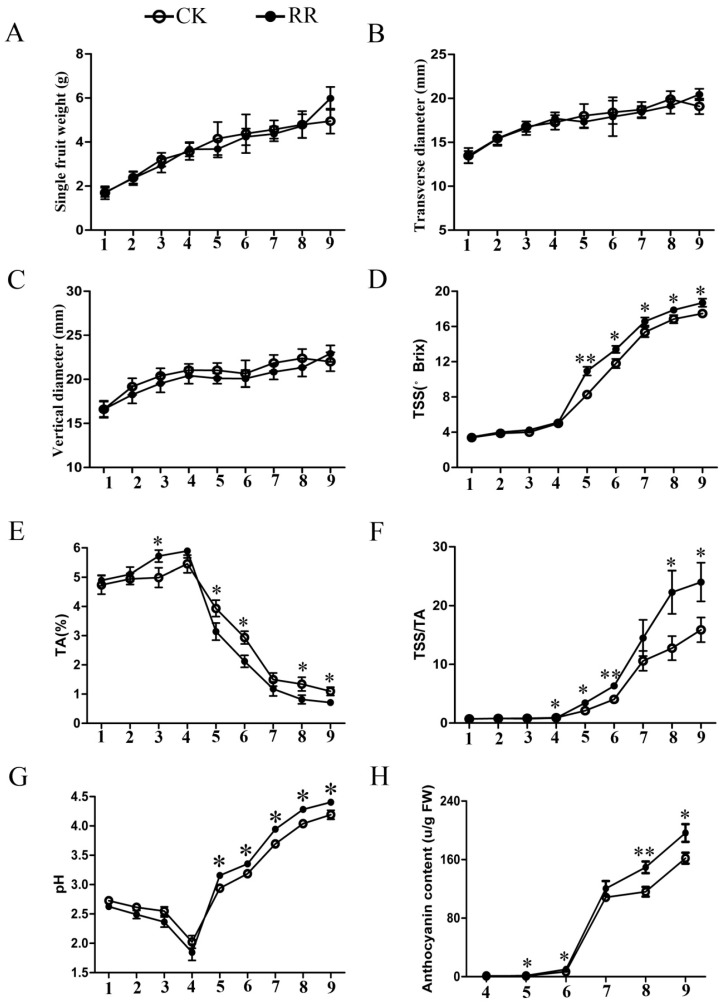
Dynamic changes of grape fruit size and contents in different growth stages. (**A**) Dynamic change of single fruit weight in different growth stages. (**B**) Dynamic change of fruit width in different growth stages. (**C**) Dynamic change of fruit vertical diameter in different growth stages. (**D**) Dynamic change of soluble solids in different growth stages. (**E**) Dynamic change of titratable acid (TA) content in different growth stages. (**F**) Dynamic change of total soluble solid/TA (TSS/TA) in different growth stages. (**G**) Dynamic change of pH in different growth stages. (**H**) Dynamic change of anthocyanin content in different growth stages. RR: Root restriction, CK: control group. 1–9 indicate that the sample collection time is 6 June, 14 June, 22 June, 4 July, 14 July, 18 July, 23 July, 1 August, and 9 August 2017, respectively. * *p* < 0.05 and ** *p* < 0.01 respectively according to two-tailed Student’s *t*-test. Error bars show the standard error between three biological replicates (*n* = 3).

**Figure 3 ijms-20-04058-f003:**
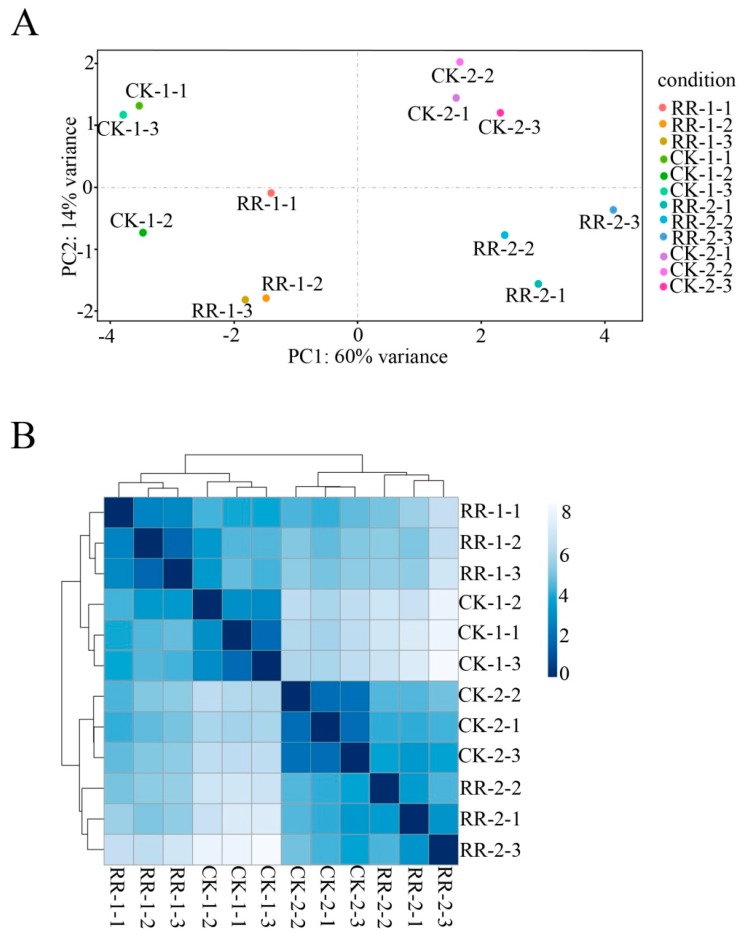
Principal component analysis (**A**) and sample to sample cluster analysis (**B**). RR-1-1, RR-1-2, and RR-1-3: Three replicas of fruit collected on 14 July 2017 under root restriction cultivation CK-1-1, CK-1-2, and CK-1-3: Three replicas of fruit in control group collected on 14 July 2017. RR-2-1, RR-2-2, and RR-2-3: Three replicas of fruit collected on 9 August 2017 under root restriction cultivation. CK-2-1, CK-2-2, and CK-2-3: Three replicas of fruit in control group collected on 9 August 2017.

**Figure 4 ijms-20-04058-f004:**
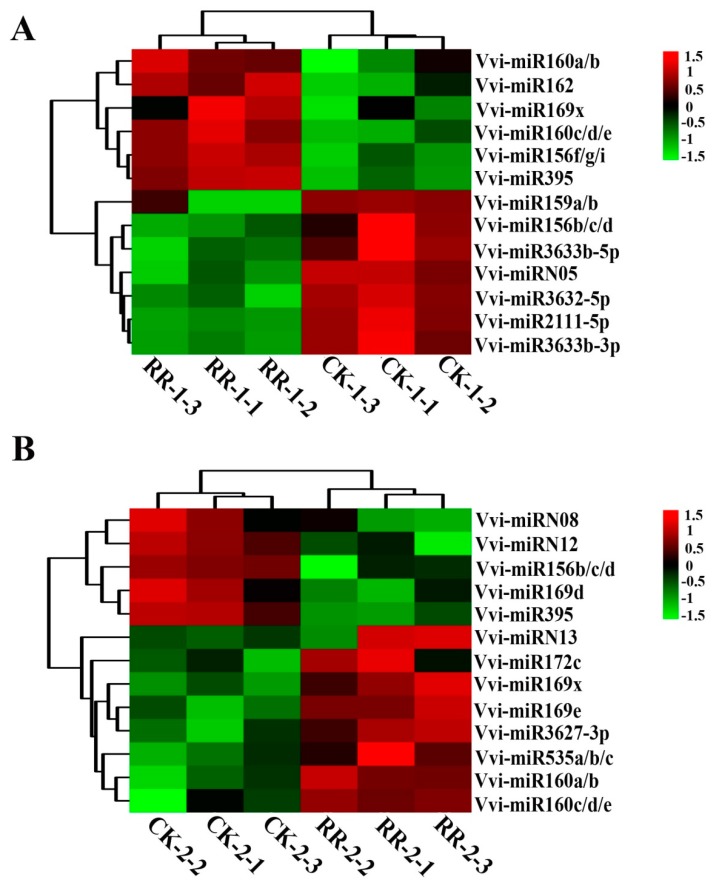
Heat maps of the normalized expression level (TPM) of differentially expressed miRNAs between RR and CK in the same period. (**A**,**B**) were from samples collected on 14 July 2017 and 9 August 2017, respectively. RR-1: fruits collected on 14 July 2017 grown under root restriction; CK-1: fruits collected on 14 July 2017 grown under non-root restriction; RR-2: fruits collected on 9 August 2017 grown under root restriction; CK-2: fruits collected on 9 August 2017 grown under non-root restriction.

**Figure 5 ijms-20-04058-f005:**
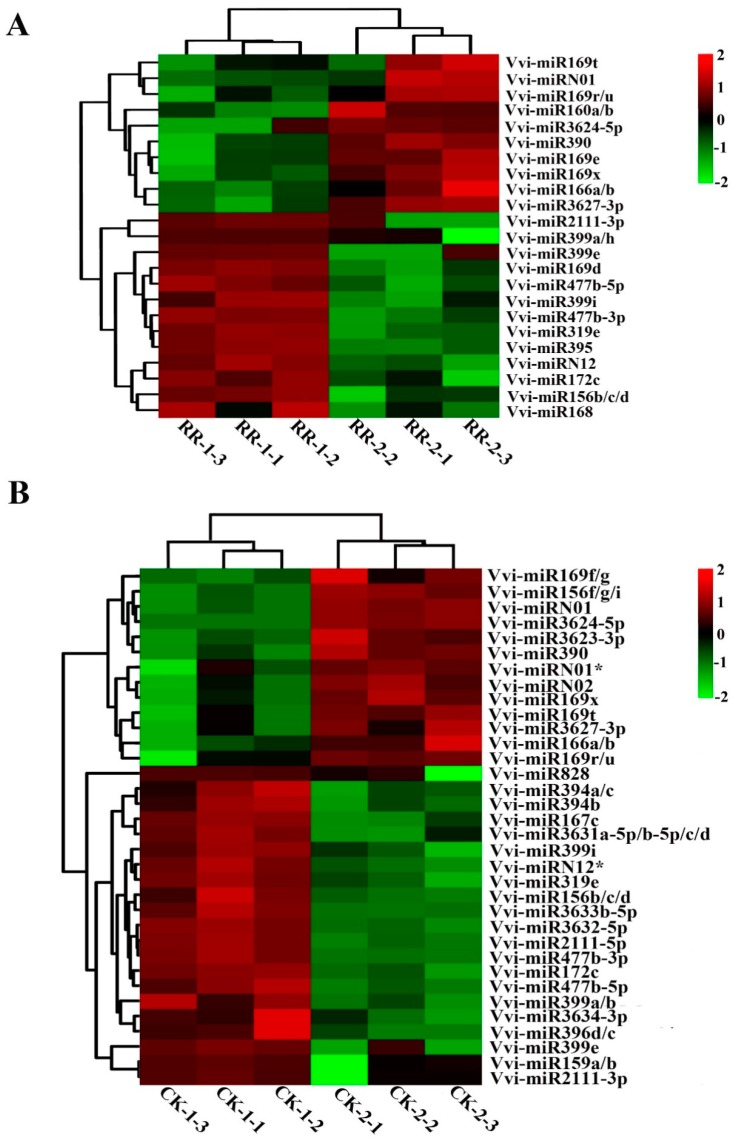
Heat maps of the normalized expression level (TPM) of differentially expressed miRNAs at different periods of the same treatment. (**A**,**B**) were from samples collected from trees grown under root restriction and non-root restriction, respectively. RR-1: fruits collected on 14 July 2017 grown under root restriction; CK-1: fruits collected on 14 July 2017 grown under non-root restriction; RR-2: fruits collected on 9 August 2017 grown under root restriction; CK-2: fruits collected on 9 August 2017 grown under non-root restriction.

**Figure 6 ijms-20-04058-f006:**
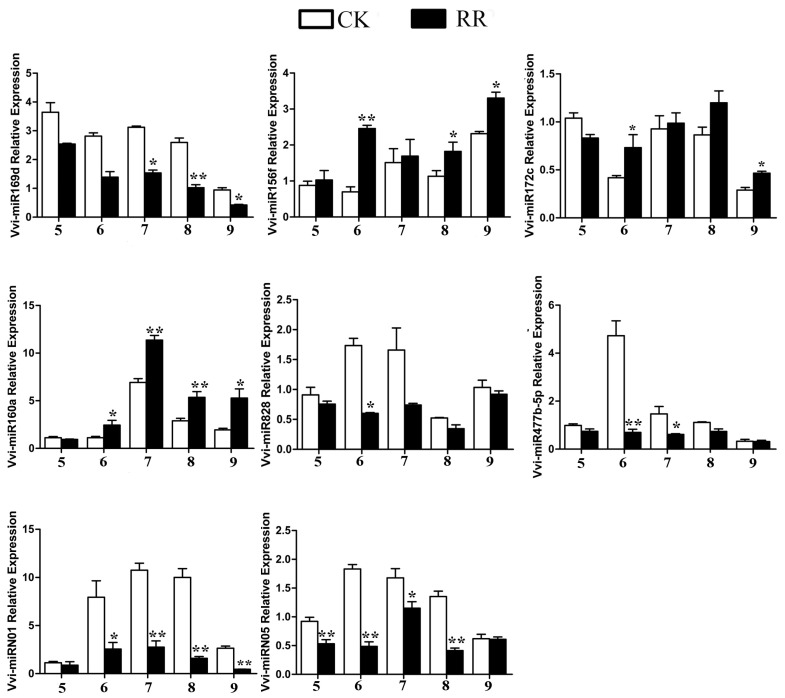
Validation and expression analysis of selected differentially expressed miRNAs derived from high throughput sequencing by tail-tailed qRT-PCR. * *p* < 0.05 and ** *p* < 0.01 respectively according to the two-tailed Student’s *t*-test. Error bars show the standard error between three biological replicates (*n* = 3). 5–9 indicate that the sample collection time is 14 July, 18 July, 23 July, 1 August, and 9 August 2017, respectively.

**Figure 7 ijms-20-04058-f007:**
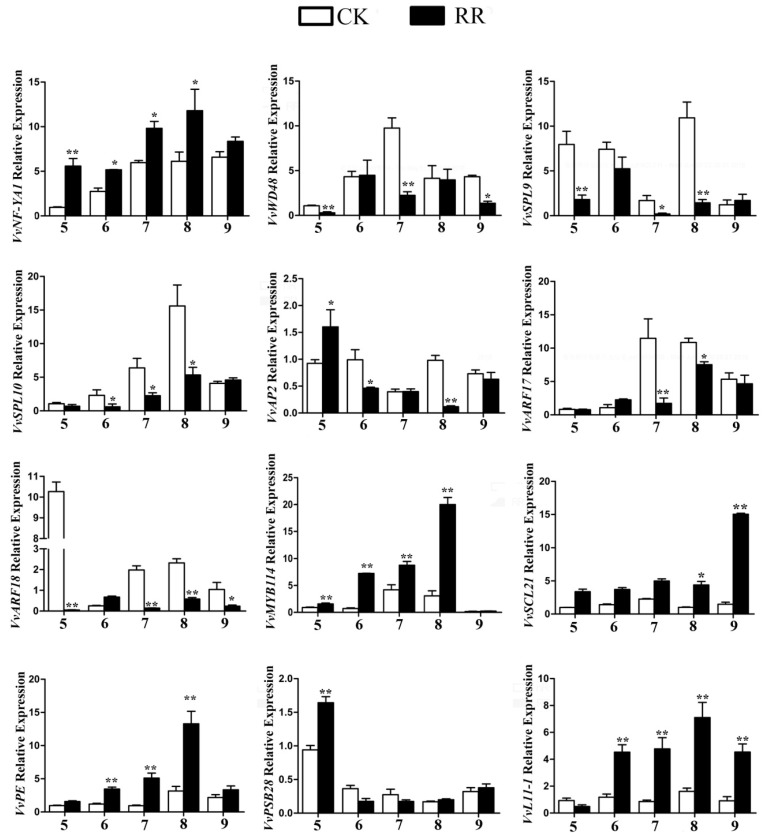
qRT-PCR analysis of target genes of selected differentially expressed miRNAs. *VvPE* (XM_002266944.4), *VvPSB28* (XM_002271630.4), and *VvL11-1* (XM_002270230.4). * *p* < 0.05 and ** *p* < 0.01, respectively, according to two-tailed Student’s *t*-test. Error bars show the standard error between three biological replicates (*n* = 3). 5–9 indicate that the sample collection time 14 July, 18 July, 23 July, 1 August, and 9 August 2017, respectively.

**Figure 8 ijms-20-04058-f008:**
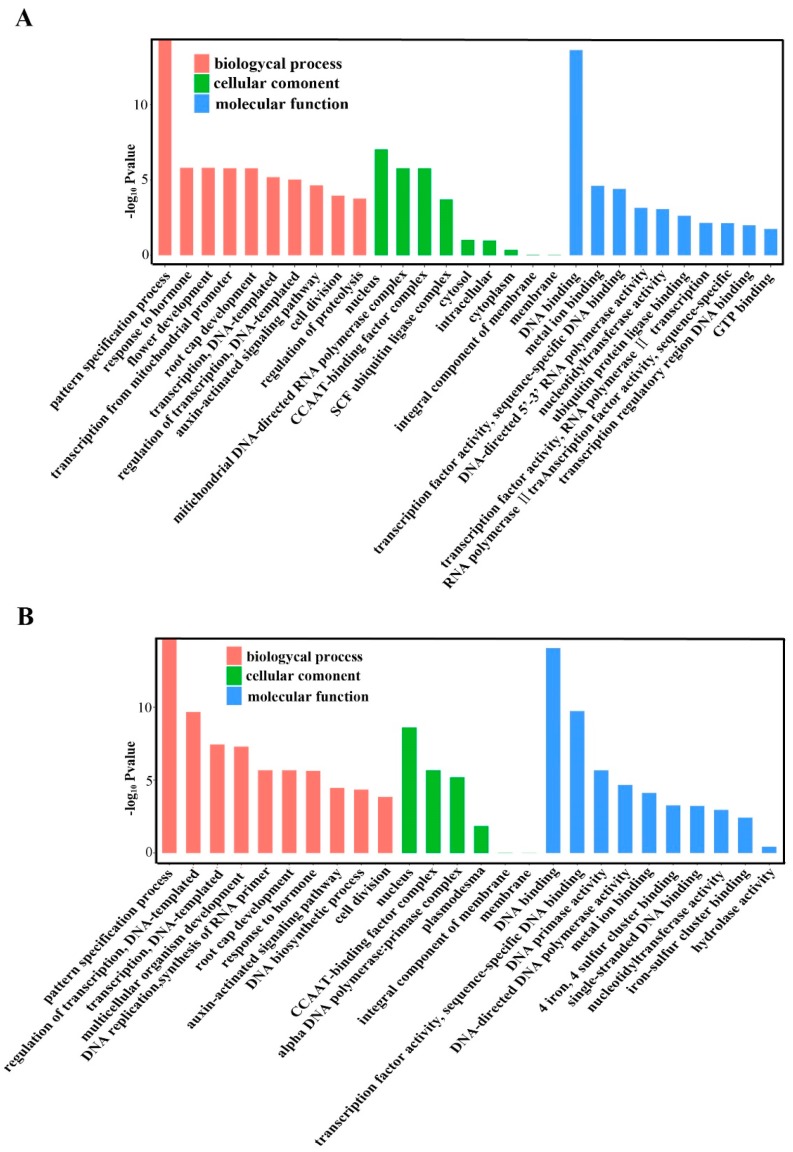
Gene ontology categories of the predicted target genes of the differentially expressed miRNAs in grape. (**A**) Gene ontology categories of the predicted target genes of differentially expressed miRNAs between RR-1 and CK-1. (**B**) Gene ontology categories of the predicted target genes of differentially expressed miRNAs between RR-2 and CK-2.

**Figure 9 ijms-20-04058-f009:**
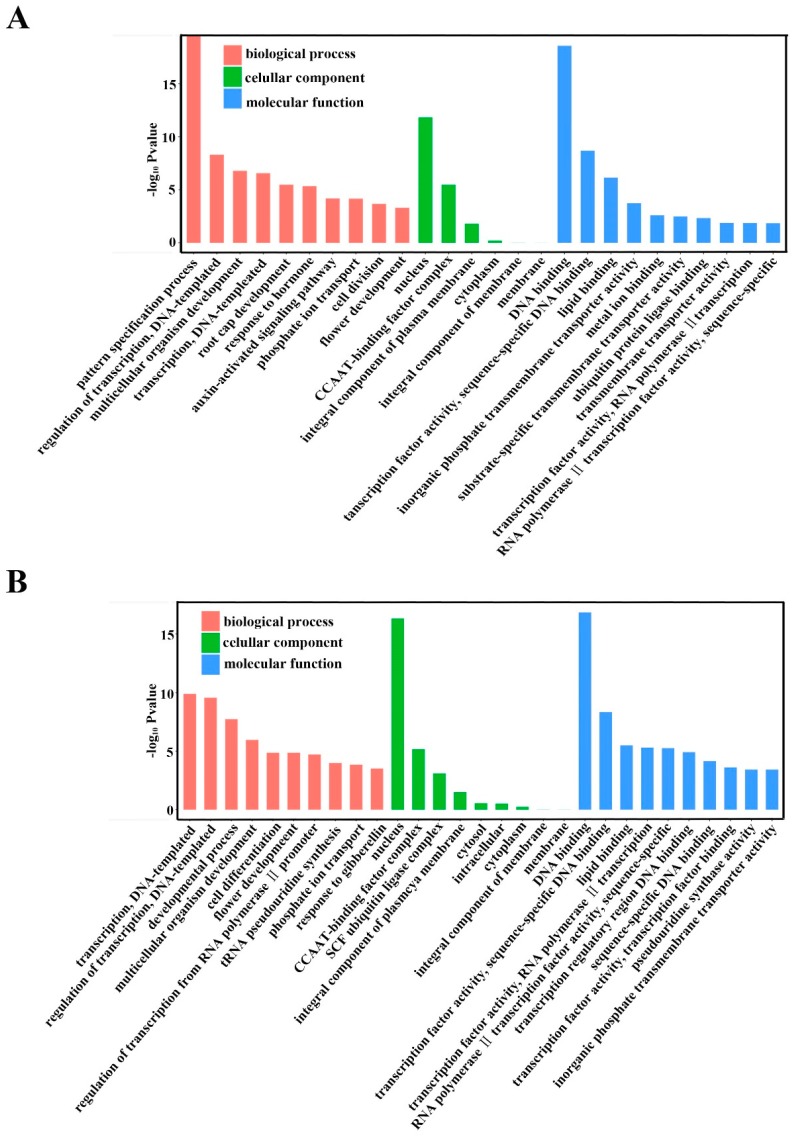
Gene ontology categories of the predicted target genes of the differentially expressed miRNAs in grape. (**A**) Gene ontology categories of the predicted target genes of differentially expressed miRNAs between RR-1 and RR-2. (**B**) Gene ontology categories of the predicted target genes of differentially expressed miRNAs between CK-1 and CK-2.

**Figure 10 ijms-20-04058-f010:**
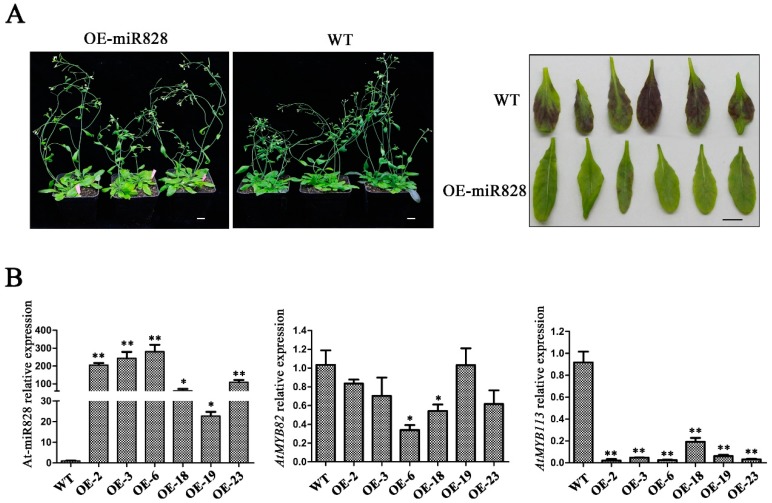
Phenotypic characterization of wild type and transgenic Arabidopsis. (**A**) Image of wild type (WT) and transgenic Arabidopsis and images of the back of rosette leaves of wild type (WT) and transgenic plants. Scale bar: 1 cm. (**B**) Expression levels of ath-miR828 detected by step-loop qRT-PCR and its target genes. * *p* < 0.05 and ** *p* < 0.01 respectively, according to two-tailed Student’s *t*-test. Error bars show the standard error between three biological replicates (*n* = 3).

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
