# Peer review of "Comparative Analysis of miRNA Abundance Revealed the Function of Vvi-miR828 in Fruit Coloring in Root Restriction Cultivation Grapevine (Vitis vinifera L.)"

_ijms, 2019, doi:10.3390/ijms20164058_

Round 1

Reviewer 1 Report

The title of the study is " Comparative analysis of miRNA abundance revealed the function of Vvi-miR828 in fruit coloring in root restriction cultivation grapevine (Vitis vinifera L.)". The aim is using miRNA expression pattern to understand the mechanism of fruit coloring. The manuscript was well written, however, there are some minor things:

Line 64, the definition of SPL9 should be clarified. Line 65-66, which are studies on grapevine? Which are they limitation? How coloring affect the fruit quality?  Please check fig. 2D and fig. 6 The statistical analysis in fig.7 should be checked.

Author Response

Comment 1: Line 64, the definition of SPL9 should be clarified.

Response: Thank you for your suggestion. We have added the full name of SPL9 gene in the manuscript (Page 2, line 65).

Comment 2: Line 65-66, which are studies on grapevine? Which are they limitation? How coloring affect the fruit quality?

Response: Thank you very much for your question. In the manuscript, we wanted to express that to our knowledge, there is no research on how root restriction cultivation influencing the growth and development of grapeberries by affecting the expression of miRNAs, especially the anthocyanin content in fruits. This is why we performed sRNA sequencing on fruits under root restriction cultivation. We are very sorry for trouble to the reviewer due to our nuclear expression. We have made changes in the manuscript (Page 2, line 67).

Comment 3: Please check fig. 2D and fig. 6. The statistical analysis in fig.7 should be checked.

Response: Thank you very much for your useful suggestions. We have carefully recalculated and analyzed the data several times. We found several errors in the significant analysis in Figure 2, Figure 6, and Figure 7. We have already corrected them. Thank you very much again for your valuable suggestions.

Reviewer 2 Report

In this article, Chen and colleagues report the effect of root restriction cultivation (RR) on miRNAs expression in grapevine. They observed that under RR, fruits presented higher soluble solids content, pH value and anthocyanin content. The analysis of sRNA sequencing data allowed to identify differentially expressed miRNAs between RR and control conditions. They also predicted and performed an annotation of their target genes. To confirm the expression data obtained by sRNA-seq, the expression of some miRNAs and their targets was verified by qPCR. Moreover, the authors overexpressed Vvi-mir828 in Arabidopsis thaliana and confirmed its involvement in anthocyanin metabolism.

The study was well designed and the article is well written. I have noted only minor shortcomings that the authors should consider to improve their manuscript:

Introduction

L40: (White et al. 2001) should be removed for reference #6.

L42: the authors should define the terms of fruit quality that they use throughout the manuscript.

L47: a dot is missing at the end of the sentence ending by “silencing [8-11]”.

L52, L53: they author should write “non-conserved” instead of “nonconserved”

L59: a space is missing before reference #34.

L65: the author should add some references to support “few studies”.

L74-75: the sentence seems odd to me.

Materials and Methods

L156: please cite tools use to filter low-quality sequences, adaptors, etc…

In general, I advise the authors to provide the version of each software.

Please provide also the parameters used to define a sequence as being of good quality.

L161: which tool is used for mapping sRNA reads on genome?

L178: psRNATarget: please specify when default parameters are used.

L186: p-value should be written like that.

L193: please provide stability values of the reference genes using a tool such as genorm or provide a reference article.

Results

L215: a space is missing between the title and the caption.

L261, L281, L286: methods elements would be better placed in the Materials and Methods part.

L270-L277: please note that even if canonical miRNAs (mainly 21 nt, mainly associated with AGO1) initiate predominantly with a 5′ U, it is not the case for other miRNAs e.g. 24-nt long miRNAs (lmiRNAs) are loaded into AGO4 proteins and start mainly with a 5′ A

(see: Mi S, Cai T, Hu Y, Chen Y, Hodges E, Ni F, Wu L, Li S, Zhou H, Long C, Chen S,

Hannon GJ, Qi Y. Sorting of small RNAs into Arabidopsis argonaute complexes is

directed by the 5' terminal nucleotide. Cell. 2008 Apr 4;133(1):116-27).

Discussion

L452: please provide the version of psRNA-Target in the Materials and Methods part.

Figure4:

The font used for the name of the miRNAs is different for Figure 4b.

Author Response

Introduction

Comment 1: L40: (White et al. 2001) should be removed for reference #6.

Response: We have removed the reference (Page 1, line 40).

Comment 2: L42: the authors should define the terms of fruit quality that they use throughout the manuscript.

Response: Thank you for your useful suggestion. We have defined the term of fruit quality in the manuscript (Page 2, line 42-43).

Comment 3: L47: a dot is missing at the end of the sentence ending by “silencing [8-11]”.

Response: We have added a dot at the end of the sentence ending by "silencing [8-11]" (Page 2, line 48).

Comment 4: L52, L53: they author should write “non-conserved” instead of “nonconserved”.

Response: We have corrected "nonconserved" to "non-cinserved" (Page 2, line 53 and line 54).

Comment 5: L59: a space is missing before reference #34.

Response: We have added a space before reference 34 (Page 2, line 60).

Comment 6: L65: the author should add some references to support “few studies”.

Response: Thank you very much for your useful suggestion. In the manuscript, we wanted to express that to our knowledge, there is no research on how root restriction cultivation influencing the growth and development of grapeberries by affecting the expression of miRNAs, especially the anthocyanin content in fruits. This is why we performed sRNA sequencing on fruits under root restriction cultivation. We are very sorry for trouble to the reviewer due to our nuclear expression. We have made changes in the manuscript (Page 2, line 67).

Comment 7: L74-75: the sentence seems odd to me.

Response: Thanks for the useful advice. What we want to express is that although there have been study on the miRNAome of the grape, there is no relevant study on how miRNA regulates the growth and development of grape fruit, especially grape under root restriction. We have rewritten the sentence (Page 2, line 72-76).

Materials and Methods

Comment 8: L156: please cite tools use to filter low-quality sequences, adaptors, etc…In general, I advise the authors to provide the version of each software. Please provide also the parameters used to define a sequence as being of good quality.

Response: Thank you for the useful suggestion. We have provide the version of the software used to filter adaptors, low-quality sequences, and so on (Page 4, line 159).

Comment 9: L161: which tool is used for mapping sRNA reads on genome?

Response: Thanks for the useful suggestion. We have provided the name of the software used for mapping sRNA reads on genome (Page 4, line 166).

Comment 10: L178: psRNATarget: please specify when default parameters are used.

Response: Thanks for the useful suggestion. We have added the default parameters (Page 4, line 184).

Comment 11: L186: p-value should be written like that.

Response: We have changed "P-value" to "p-value" in the manuscript.

Comment 12: L193: please provide stability values of the reference genes using a tool such as genorm or provide a reference article.

Response: Thanks for the useful suggestion. We have added the reference in the manuscript (Page 5, line 198).

Results

Comment 13: L215: a space is missing between the title and the caption.

Response: We have added a space between the title and the caption (Page 5, line 219).

Comment 14: L261, L281, L286: methods elements would be better placed in the Materials and Methods part.

Response: Thanks for the useful suggestions. We have rewritten the Materials and Methods part following the suggestion (Page 4, line 162-163, line 165-166 and line173 and page 9 line 266, line 285 and line 290).

Comment 15: L270-L277: please note that even if canonical miRNAs (mainly 21 nt, mainly associated with AGO1) initiate predominantly with a 5′ U, it is not the case for other miRNAs e.g. 24-nt long miRNAs (lmiRNAs) are loaded into AGO4 proteins and start mainly with a 5′ A (see: Mi S, Cai T, Hu Y, Chen Y, Hodges E, Ni F, Wu L, Li S, Zhou H, Long C, Chen S, Hannon GJ, Qi Y. Sorting of small RNAs into Arabidopsis argonaute complexes is directed by the 5' terminal nucleotide. Cell. 2008 Apr 4;133(1):116-27).

Response: Thanks for the useful propose. We have modified the description and added this reference in the manuscript (Page 9, line 281-283).

Discussion

Comment 16: L452: please provide the version of psRNA-Target in the Materials and Methods part.

Response: Thank you for the useful propose. We have added the version of psRNA-Target in the Material and Methods (Page 4, line 183)

Comment 17: Figure4: The font used for the name of the miRNAs is different for Figure 4b.

Response: We have made changes to the font in Figure 4B.